# Genomic Risk Factors for Cervical Cancer

**DOI:** 10.3390/cancers13205137

**Published:** 2021-10-13

**Authors:** Dhanya Ramachandran, Thilo Dörk

**Affiliations:** Gynaecology Research Unit, Department of Gynaecology and Obstetrics, Comprehensive Cancer Center, Hannover Medical School, D-30625 Hannover, Germany; Ramachandran.Dhanya@mh-hannover.de

**Keywords:** cervical cancer, cervical dysplasia, HPV, HLA, GWAS, genetic susceptibility, meta-analysis, SNP, eQTL, papillomavirus

## Abstract

**Simple Summary:**

Cervical cancer is the fourth leading cause of cancer-related deaths in females worldwide. Despite an active immunisation strategy in many countries, high-risk human papillomavirus infection and environmental factors may result in cervical cancer progression. Genetic factors contributing to host–pathogen interactions are incompletely understood and remain largely unknown, apart from the variants at the human leukocyte antigen (HLA) locus on 6p21.3, which replicates across populations. Novel cervical cancer susceptibility loci from recent large biobank-based genome-wide association studies and meta-analyses are likely to be more robust as compared to previous candidate gene based studies. In this review, we summarize findings from recent genome-wide association studies on cervical cancer and propose variants that may be universal susceptibility loci.

**Abstract:**

Cervical cancer is the fourth common cancer amongst women worldwide. Infection by high-risk human papilloma virus is necessary in most cases, but not sufficient to develop invasive cervical cancer. Despite a predicted genetic heritability in the range of other gynaecological cancers, only few genomic susceptibility loci have been identified thus far. Various case-control association studies have found corroborative evidence for several independent risk variants at the 6p21.3 locus (HLA), while many reports of associations with variants outside the HLA region remain to be validated in other cohorts. Here, we review cervical cancer susceptibility variants arising from recent genome-wide association studies and meta-analysis in large cohorts and propose 2q14 (*PAX8*), 17q12 (*GSDMB*), and 5p15.33 (*CLPTM1L*) as consistently replicated non-HLA cervical cancer susceptibility loci. We further discuss the available evidence for these loci, knowledge gaps, future perspectives, and the potential impact of these findings on precision medicine strategies to combat cervical cancer.

## 1. Epidemiology and Heritability of Cervical Cancer

### 1.1. Risk and Prevention

Cervical cancer is one of the leading causes of cancer-related deaths in women worldwide [1,2,3]. Infection by known high risk sub-types of human papillomavirus (HPV 16, 18, 31, 33, 34, 35, 39, 45, 51, 52, 56, 58, 66, 68, and 70) as well as environmental or life style factors such as multiple sexual partners, smoking, obesity, high parity, early menopause, co-infection with *Chlamydia trachomatis*, herpes simplex virus type-2 (HSV2), or human immunodeficiency virus (HIV) are known to increase the risk of developing cancer [4,5,6,7,8,9]. Some of the epidemiological factors may correlate with specific cervical cancer histology groups such as adenocarcinomas or squamous carcinomas [10].

As preventive measures, screening at regular intervals at clinics or self-testing at home, followed by Papanicolaou “pap” smear testing, or PCR based tests for HPV typing, help to identify the early stages of infection [11,12]. Vaccination against the most prevalent high risk HPV types (Cervarix, Gardasil, or Gardasil 9) is available in many developed nations [12,13,14]. Treatment options include surgery, irradiation, chemotherapy, targeted immunotherapy, or combinations thereof. Despite this, the mortality rate from cervical cancer is still high in developing countries.

### 1.2. HPV-Associated Pathogenesis

The lower part of the uterus, called the *cervix uteri,* contains an inner and an outer compartment (endocervix and ectocervix). The thin, flat squamous cells lining the ectocervix meet the columnar glandular cells of the inner endocervix at the transformation zone, which is prone to pre-cancerous lesions caused by HPV. The first infections in low grade squamous intraepithelial lesions (LSIL), or cervical intraepithelial neoplasia stage 1 (CIN1), often resolve on their own (in 90% cases up to two years) [6]. Re-infection may occur multiple times, without leading to cancer development in most cases. Only a small fraction of women go on to develop invasive cervical cancer after HPV infection, indicating that genetic variability may play a vital role in resolving infection and preventing progression to invasive disease [15]. A persistent HPV infection over time, gradual worsening of the lesions, viral load, tissue composition at the site of viral integration, and multiple reinfections are associated with the development of high grade squamous intraepithelial lesion (HSIL), or cervical intraepithelial neoplasia stage 2 and 3 (CIN2 and CIN3) and further, cancer in situ (CIS) [16]. These stages are together termed as high-grade dysplasia and can progress into carcinoma or invasive cervical cancer [6,17,18] (Figure 1).

Although all high-risk human papillomaviruses predispose a woman to cervical cancer, there are notable differences between them. The most prevalent anogenital HPV type, HPV16, is also the most carcinogenic. Variations within the viral genome appear to modulate the pathogenicity of the respective lineages of HPV16, HPV33, or HPV45 [19,20,21]. Such genomic evolution of the virus may be partially driven by host cell DNA-editing enzymes of the APOBEC family [22]. The mechanisms underlying the different pathogenicity of diverse lineages are largely unknown, but may relate to differential expression levels or alternate splicing of viral oncogenes [23]. Further differences have been noted with regard to the tumour characteristics. The second most carcinogenic HPV type, HPV18, accounts for approximately 12% of squamous cell carcinoma, but 37% of adenocarcinoma of the cervix worldwide [24], suggesting etiological differences in tumour development after HPV16 or HPV18 infections. Patterns of integration within the host genome can also vary between different HPV types. HPV18 appears to integrate fully in cervical intraepithelial neoplasia grade 3 (CIN3) or invasive cancer whereas HPV16 can be either completely integrated or remain as episomes [25,26,27]. HPV integration sites within the human genome have been closely scrutinised with reported changes in gene and protein expression near integration hot spots as well as differential pathway activation [27,28,29]. Possible mechanisms include differences in promoter methylation, which have also been reported in HPV positive versus negative lesions [30,31,32], direct disruption of genes, or activation of retroelements [33].

The viral DNA replicates inside the host cell and starts expressing early genes *E6* and *E7*, among others [7]. The E6 protein is known to bind and initiate the degradation of p53, a well-characterised tumour suppressor, blocking apoptosis and accumulating DNA damage within the cells, which can lead to uncontrolled proliferation [34,35,36]. The E7 protein binds and inactivates the retinoblastoma (Rb) protein, which otherwise sequesters the transcription factor E2F [37,38,39]. Upon the inactivation of Rb, E2F lifts the cell cycle checkpoint inhibition, and induces unchecked cell proliferation. E7 via E2F also upregulates cyclin-dependent kinase inhibitor 2A (p16^INK4A^, or CDKN2A) expression, which acts as a prognostic biomarker for cervical cancer [40,41,42]. *CDKN2A* is then gets hypermethylated [43,44,45], however, this inhibition of *CDKN2A* cannot effectively arrest the cell cycle while Rb is blocked and p53 is degraded. E7 also suppresses p53 effector and tumour suppressor protein cyclin-dependent kinase inhibitor 1A (p21 or CDKN1A) [37,38,39].

E6 and E7 overexpression appears to downregulate the toll-like receptor (TLR) mediated type I interferon (IFN- α and ß) response [46]. Additionally, once the HPV integrates inside the cells, this interferon mediated response is ineffective [47]. Keratinocytes express cytokines, which are crucial for the activation of macrophages and Langerhans cells and for T-cell activation, but HPV episomes have been shown to downregulate the cytokines IL-1ß and IL-6 [48,49]. In some women, due to a combination of such mechanisms, immune cells are not able to constrain viral replication, and the unrestrained cell proliferation due to the E6 and E7 downstream activities results in progression to HSIL and ultimately, cervical cancer (Figure 1).

HPV-induced inactivation of p53 and unscheduled replication and cell proliferation make the host cell prone to mutagenesis. As with all cancers, the host genome acquires selective advantage through the accumulation of somatic mutations and epigenetic changes. Some 537 genes associated with cervical cancer progression have been included in an early manually curated cervical cancer gene database [50], while genes with an experimentally validated role in cervical cancer have been listed in the Disease-Gene Association database DisGeNET [51]. More recently, large sequencing efforts such as those of the TCGA and the BioRAIDs consortia have established key genes and pathways for cervical cancer [27,52]. It is possible, but yet under-investigated, that differential regulation of some of them are also relevant in the germline predisposition of cervical cancer risk.

### 1.3. Heritability of Cervical Cancer

Genetic factors contributing to the development of HSIL and invasive cervical cancer are largely unknown. However, some evidence for familial clustering has been noted for more than 60 years [53], thus previous studies have examined whether there is a hereditary component to acquiring cervical cancer. There is clear evidence for a high familial relative risk (FRR), which has been deduced from the Swedish cancer registry and indicated towards disease clustering in families [54,55,56]. These studies reported that female offspring and siblings have a relative risk (RR) of 1.5–2.3 for developing cervical cancer. This risk is substantial and comes close to the familial relative risk for breast cancer, which has a strong genetic component.

However, in contrast to breast and ovarian cancer, there have not been many large multiple-case families described with cervical cancer [53,55,57,58,59], suggesting that high-penetrance germline variants are rare in this malignancy and many of the heritable factors may be low- or intermediate penetrant and may act in synergy with HPV infection. With regard to the outstanding role of HPV infection, cervical cancer has sometimes been called a unique human neoplasia that has one single necessary cause [60]. Nevertheless, there has been accumulating evidence that cervical cancer can occur in the absence of HPV [61,62,63] and these tumours with undetected HPV have specific molecular pathology [27,64]. It is important to note that some of the HPV negative cervical cancer cases may arise from false diagnoses or outdated genotyping methods [65]. However, with the advancements in HPV genotyping and enhanced detection of more HPV sub-types, it has become possible to identify true HPV-negative cervical cancer cases. Genome-wide sequencing of a large series of such cases may reveal whether germline variants in certain genes are enriched in cancers with undetected HPV.

Further studies towards the hypothesis of germline predisposition suggested that heritability via genetic factors might contribute some 27–36% of cervical cancer risk variation [56,66], and a large proportion of this heritability was attributed to unidentified autosomal common SNPs with low penetrance [67]. A pan-cancer heritability study using UK Biobank and US Kaiser Permanente Genetic Epidemiology Research on Adult Health and Aging (GERA) data found an array heritability estimate of 7% for cervical cancer, which was similar to the estimates for ovarian or colon cancer [68]. As will be discussed in Section 2.2 below, several genomic loci have been identified by recent GWAS at genome-wide significance. However, the contribution of these variants is small, and the major fraction of the estimated heritability still remains to be defined.

### 1.4. Candidate Gene Based Studies

There have been many candidate-gene based studies performed for cervical cancer, but the findings have been restricted to specific populations. Since host genetic factors are thought to play a major role in the response to cancer and HPV infection, most cervical cancer candidate gene based studies have focused on genes with relevant roles in immunity or carcinogenesis.

Candidate cervical cancer susceptibility gene variants have been reported in the tumour-suppressor gene TP53 [69,70,71] or the p53 regulating ubiquitin ligase gene MDM2 [70,72,73], and in further DNA damage response or cell cycle genes such as ATM [74], BRIP1 [75], CDKN1A [76,77,78], CDKN2A [79], FANCA, FANCC, and FANCL [80], XRCC1 [81,82,83], or XRCC3 [84]. Variants in immune response genes, which may confer immune advantage to the virus or to the host, in genes such as T-cell surface molecules CD83 [85,86] and CTLA4 [87], CARD8 [88], or secreted factors such as tumour necrosis factor alpha (TNFA) [89,90,91,92], interleukins [93,94,95,96], transforming-growth factor beta (TGFB1) [97], interferon-gamma (IFNG) [76,98] have also been studied, among many others.

Despite these considerable efforts, the vast majority of proposed risk variants from candidate gene studies have not been replicated (e.g., a debated ArgR72Pro variant in p53 [99]) and have not reached statistical significance in large case-control studies or meta-analyses (except for certain HLA alleles, e.g., [67]). With technological advancements over the past decade, stronger evidence for additional risk variants has come from the massively parallel analysis of millions of variants throughout the whole genome. In the following section, we will discuss the progress made through these genome-wide association studies.

## 2. Genomic Susceptibility Variants for Cervical Cancer

### 2.1. Genome-Wide Association Studies

GWAS are powerful tools to identify common susceptibility variants in the population and have very successfully been applied to cancer research [100]. After genotyping and imputation, association analysis is performed using software such as PLINK or Regenie [101,102]. After associated variants are identified, replication studies in additional cohorts and meta-analysis are performed to validate new loci. Fine-mapping approaches along with bioinformatic annotations and colocalisation help to identify the causal SNP from independent sets of correlated, highly associated variants (iCHAVs). In silico predictions are used to annotate variants for known chromatin marks, genes in the vicinity, and for testing gene and pathway enrichment. These predictions become crucial in understanding the role of identified susceptibility variants since a majority of them are from the non-coding genome [103,104]. Furthermore, functional assays are designed to assess biological functions of the lead variants in the form of luciferase reporter assays, quantitative trait loci for expression, methylation, splicing, and protein levels (eQTL, metQTL, sQTL, and pQTL), chromatin immunoprecipitation (ChIP), chromosome conformation capture and related technologies (3C, 4C, 5C, Hi-C, ChIA-PET), or functional studies after genome editing of the sequences containing the variant by CRISPR/Cas or related techniques [105,106] (Figure 2).

GWAS are conducted to identify common trait-associated variants above the genome-wide significance (GWS) threshold of *p* < 5 × 10^−8^, however, sub-significant variants may contribute small effects towards traits and are worthy of scrutiny [106]. The availability of rich data sources such as GTEx, HaploReg, SNPNexus, SNiPA, among many others, enable lookup of known eQTL and chromatin status, motif changes, linked variants, and previous known associations [107,108,109,110]. Similarly, the availability of summary statistics from published GWAS studies enables meta-analyses and validation of proposed trait-associated variants in different populations. These may eventually aid previously sub-significant variants to now cross the GWS threshold since combined studies have an increased power to detect common variants with small effects.

GWAS in large cohorts derived from biobanks (UK, FinnGen, Japan, Estonia, IARC, among others) are increasingly being used to quantify disease risk, derive polygenic risk scores (PRS), determine the genetic correlation between traits having shared environmental factors, and test causality between exposures and outcomes (Mendelian randomisation) [111,112,113]. These will drive future decisions in precision medicine and preventive screening [109,114].

### 2.2. Results from Cervical Cancer GWAS

The power of GWAS for the detection of cervical cancer susceptibility has been increasingly exploited over the past decade [115]. There have been a handful of cervical cancer-specific GWAS worldwide, which, however, have been complemented with recent studies from large biobank based cohorts (Table 1). Some studies have focused on invasive cervical cancer while others combined dysplasia and invasive cancer or have analysed dysplasia separately. The features and main findings of these GWAS are sequentially summarised in Table 1.

Although the GWAS addressed different populations, there has been some overlap between the results. We will consider this shared evidence across two or more GWAS as successful replication. Most of the consistent genome-wide significant variants arose from the human leukocyte antigen (HLA) locus in the chromosome 6p21.3 region. However, three non-HLA signals on chromosomes 2q13 (*PAX8*), 5p15.33 (*TERT-CLPTM1L*), and 17q12 (*GSDMB*) have also been replicated in different study populations. In the following sub-sections, we highlight these consistent loci, but also pay attention to those that still need replication in independent cohorts.

#### 2.2.1. 6p21.3 (HLA)

The human leukocyte antigen (HLA) locus on chromosome 6p21.3 is the human equivalent of the major histocompatibility complex (MHC) and contains several genes that encode membrane proteins responsible for the regulation of the immune system [125,126,127]. The MHC is divided into three subclasses: the class I region, which includes the classical, highly polymorphic *HLA-A*, *HLA-B*, and *HLA-C* genes, and the less polymorphic *HLA-E*, *HLA-F*, and *HLA-G* genes as well as the non-canonical MHC class I-related chain (MIC) genes *MICA* and *MICB* that encode ligands for the activating natural killer cell receptor NKG2D; the class II region, which includes the *HLA-DPA1*, *HLA-DPB1*, *HLA-DQA1*, *HLA-DQA2*, *HLA-DQB1*, *HLA-DQB2*, *HLA-DRA*, *HLA-DRB1*, *HLA-DRB2*, *HLA-DRB3*, *HLA-DRB4*, and *HLA-DRB5* genes as well as less variable genes involved in antigen processing and presentation; and the class III region, which contains genes implicated in inflammatory responses, leukocyte maturation, and the complement cascade [127]. Class I and Class II MHCs form two types of peptide presenting complexes, and differential representation or peptide binding of these may be associated with differential disease susceptibility.

Although we highlight the HLA region as the first consistent cervical cancer susceptibility locus, it is, in fact, made up of several separate signals. The first cervical cancer GWAS, performed in the Swedish population, identified multiple variants at the HLA locus [116]. It confirmed allelic associations with *HLA-B**07:02, *HLA-DRB1**13:01-*DQA1**01:03-*DQB1**06:03, and *HLA-DRB1**15:01-*DQB1**06:02, which had previously been reported in candidate gene studies, and further identified three novel loci for CIN3 in the MHC region: rs9272143 between *HLA-DRB1* and *HLA-DQA1*; rs2516448 adjacent to *MICA*; and rs3117027 at *HLA-DPB2* [115,116]. Interestingly, the risk allele rs2516448 was in perfect linkage disequilibrium with a frameshift mutation (the A5.1 allele) in exon 5 of *MICA*, resulting in a truncated MICA protein and less membrane-detectable MICA in cervical lesions, which may compromise the immune response towards HPV infection or neoplastic change [115,116,128,129]. Other polymorphisms at the same *MICA* exon 5 microsatellite sequence were also associated with cervical cancer [128]. Further SNPs in the vicinity (rs9271898, rs3130196, and rs73730372) were identified by follow-up investigations by combining cohorts and via pathway analysis by the same group [115,118,130].

There were several replications of these findings. The first Asian cervical cancer GWAS replicated the HLA locus in identifying another signal (rs4282438, *HLA-DPB2*) in the Chinese population [117]. Apart from a multi-centric study on Caucasians, which corroborated variants at the HLA locus (esp. rs9271858) [131], a cervical cancer GWAS meta-analysis combining >400,000 samples from the UK Biobank and Kaiser Permanente GERA cohorts also confirmed previously known variants at the HLA locus and identified a novel HLA signal, rs2856437 at *PBX2* [68]. The UK Biobank cervical cancer GWAS, which combined CIN3 and invasive cervical cancer, confirmed variants at *HLA-DQA1* (rs9272050), *MICA* (rs6938453), and *HLA-DQB1* (rs55986091), of which *HLA-DQA1* (rs9272050) was also replicated at a genome-wide significance in the FinnGen biobank cohort [121]. The *MICA* variant rs6938453 is only very weakly correlated with the initially reported variant rs2516448 (r^2^ = 0.22). This study furthermore identified a novel association with rs9266183 in *HLA-B* that encodes a rare missense substitution, p.Asp54Gly [121].

Although some studies have focused on invasive cervical cancer or did not distinguish between invasive cancers and dysplasia, there is solid evidence that HLA variants already affect the risk at the dysplasia stage [116,121,122,132]. The first Swedish GWAS was performed on high-grade dysplasia, CIN3 [116]. The UK Biobank and FinnGen study also performed a GWAS restricted to cervical dysplasia and reported rs9272245 (*HLA-DQA1)* as a signal for dysplasia alone [121]. A recent trans-ethnic GWAS meta-analysis including the Estonian population proposed two signals at the HLA locus, rs1053726 (*HLA-B*) and rs36214159 (*HLA-DQA1*), from a specific analysis that only included dysplasia cases [122]. This indicates that the risk conferred by at least some *HLA* alleles manifests early in the process of cervical carcinogenesis.

Chen et al. investigated HLA alleles specifically and identified significant associations with cervical dysplasia and cancer for *HLA-B**07:02, *HLA-B**15:01, *HLA-DRB1**13:01, *HLA-DRB1**15:01, *HLA-DQA1**01:03, *HLA-DQB1**06:03, *HLA-DQB1**06:02, and *HLA-C**07:02 in the Swedish population [133]. Further studies in different populations have reported, amongst others, associations with *HLA-DQB1**05:01 and *HLA-DRB3**99:01, which have recently been recovered in the UK Biobank [134,135,136]. More work will be required to reduce the reported associations in the HLA region to conditionally independent alleles with high-confidence associations.

From the known cervical cancer GWAS thus far, ten apparently independent HLA signals have emerged that are not in linkage disequilibrium at r^2^ > 0.3, and another five HLA signals have been suggested from the cross-cancer analyses (Table 1). As the pattern of linkage disequilibrium at the *HLA* region is complex and varies according to ethnic background, there could be hidden correlation for some of the GWAS hits. Of the clearly independent hits, the two signals initially represented by rs2516448 and rs9272143 have been replicated by three or four other GWAS, respectively, as well as through direct genotyping in case-control studies [131,132]. The genes underlying these consistent signals in the HLA region are unknown, although plausible candidates including *MICA* and *HLA-DRB1* have been proposed [128,132]. A study of 278 affected sib pairs revealed significant excess genetic sharing for all three HLA class II loci studied: *HLA-DPB1*, *HLA-DQB1*, and *HLA-DRB1* (with the strongest evidence for DQB1 and DRB1) whereas no evidence of excess sharing was observed for the HLA class I *HLA-A* and *HLA-B* loci [137]. On the other hand, *HLA-B* has also been repeatedly identified as a separate GWAS locus [121,122]. As the HLA region is very large and gene-dense, several genes may be affected and the causal genes may not necessarily be those in the closest vicinity. Some evidence has indicated that there may be long-range transcriptional regulation of HLA gene expression in cervical specimens [132]. Adding to the complexity, such regulations might be dependent on HPV status and on tissue constitution.

#### 2.2.2. 17q12 (GSDMB)

The first GWA study in the Chinese population identified two novel risk loci at 4q12 (rs13117307, *EXOC1*) and 17q12 (rs8067378, *GSDMB*) [117]. While the first one has not been fully replicated, the signal on 17q12 also came up in a transethnic meta-analysis GWAS from the Estonian Biobank, the FinnGen study, the UK Biobank, and Biobank Japan (rs12603332, [122]). The closest gene is *GSDMB*, which encodes a member of the gasdermin family that forms pores in the membrane to aid the release of cytokines. Gasdermin B participates in the regulation of cell pyroptosis, a pro-inflammatory form of regulated cell death that is designed to attract a nonspecific innate response to the site of infection [138,139]. It also appears to activate STAT3 signalling and thereby tumour growth [140]. Interestingly, higher levels of Gasdermin B (also known as Gasdermin-like, GSDML) had previously been associated with tumour progression in cervical cancer [141]. Although it is an attractive candidate for cervical cancer risk, experimental evidence for *GSDMB* as the causal gene underlying the signal on 17q12 is still lacking.

#### 2.2.3. 2q13 (PAX8)

A cervical cancer GWAS meta-analysis combining >460,000 samples from the UK Biobank and the Kaiser Permanente Genetic Epidemiology Research on Adult Health and Ageing investigated heritability and pleiotropy across 18 cancer types [68]. This study not only confirmed previously known variants at the HLA locus, but also identified a novel variant at genome-wide significance on 2q13 (rs10175462) near the *PAX8* gene that encodes the transcription factor Paired box 8 [68]. This signal replicated in the UK Biobank and in the FinnGen Biobank [121] as well as in a German case-control study [142] and in the transethnic meta-analysis GWAS from the Estonian Biobank, the FinnGen study, the UK Biobank and Biobank Japan (rs4849177, [122]). Two weakly correlated variants, rs4848320 and rs1110839, had previously been associated with cervical cancer in a candidate gene study of *PAX8-AS1* haplotypes [143]. In gene expression analyses, the variant rs10175462 presented as a cis-eQTL for the *PAX8-AS1* non-coding RNA and also appeared to modulate *PAX8* levels in response to HPV [142]. *PAX8* is also emerging as an important transcription factor for other gynaecological cancers [144] but more work will be required to elucidate its role in cervical cancer pathogenesis. It is possible that the *PAX8*/*PAX8-AS1* locus has a central role in cervical biology and pathology, as it has also been associated with cervical ectropion and cervicitis [122].

#### 2.2.4. 5p15.33 (CLPTM1L)

The UK Biobank cervical cancer GWAS identified a genome-wide significant locus at *CLPTM1L* (rs27069) [121]. This finding was replicated in an independent series from the FinnGen biobank cohort at *p* = 2.5 × 10^−7^ [121] and remained genome-wide significant in a trans-ethnic meta-analysis that included the UK, Finnish, Japanese, and Estonian Biobanks [122]. rs27069 is located 10 kbp upstream of *CLPTM1L*, which encodes a membrane protein involved in apoptosis. Some 50 kbp apart lies *TERT*, the gene encoding human telomerase reverse transcriptase. The *CLPTM1L-TERT* locus is relevant for several gynaecological cancers [121,145,146,147,148], however, the functional significance of the identified variant and its contribution to cervical cancer pathogenesis via the *CLPTM1L-TERT* locus remain to be elucidated.

### 2.3. Other GWAS Loci for Cervical Cancer or Dysplasia

The first cervical cancer GWAS in the Japanese population failed to identify any SNPs at genome-wide significance, but a second, larger GWAS in the East Asian population (with cohorts from China and Japan), identified and linked their top novel variant rs59661306 on chromosome 5q to the *ARRDC3* (Arrestin domain-containing 3) gene, encoding a known tumour suppressor and regulator of insulin sensitivity [119,149,150,151]. However, these results have not yet been replicated in European populations.

Another study on cross-trait analysis in gynaecological cancers with some overlap with the Japanese Biobank identified two novel cervical cancer variants at *INS-IGF2* (rs150806792) and *SOX9* (rs140991990), requiring further validation [152]. The most recent Japanese Biobank cervical cancer GWAS did not identify any new variants at GWS, although it confirmed previous findings [120] (Table 1).

Cross-trait analyses have also been performed in the pan-cancer meta-analysis of the UK Biobank and Kaiser Permanente GERA cohorts [68]. Although this study did not find evidence for a genetic correlation of cervical cancer with any other cancer at *p* < 0.05, it identified some loci with pleiotropic variants that reached genome-wide significance after combining cervical cancer with other cancer cases. Among those were seven HLA variants but also two signals at 4q24 (rs10007915, *TET2*) and 8q24.21 (rs117952826, *CASC8*) (Table 1). Further work will be required to validate whether these associations were significantly driven by cervical cancer.

Bowden et al. performed a separate GWAS for invasive cervical cancer in the UK and FinnGen Biobanks and reported two novel signals for invasive cancer only, rs138446575 (*TTC34*) and rs117960705 (*ACACB*) [121]. Polymorphisms in the latter gene have also been associated with obesity and diabetes [153]. However, these results would need to be replicated and the causal genes identified.

The first GWAS in the Chinese population identified a risk locus on chromosome 4q12 (rs13117307, *EXOC1*) [117], which has not been replicated by subsequent GWAS results, although a multicentric case-control study reported borderline evidence of association [131]. A second GWAS designed to detect association with neoadjuvant chemotherapy response discovered four loci for prognosis at 4q (rs6812281), 10q (rs4590782), 14q (rs1742101), and 16q (rs1364121) [154]. These loci might affect treatment response rather than the primary risk and have not been validated thus far.

Koel et al. performed a GWAS trans-ethnic meta-analyses that included Estonian, UK, Finnish, and Japanese Biobanks, and reported genome-wide significant signals for five loci associated with invasive cervical cancer including one novel signal at *LINC00339/CDC42* (rs2268177) on chromosome 1p36 [122]. Furthermore, this study reported a novel locus for cervical dysplasia at *DAPL1* (rs12611652) on chromosome 2q24. In a joint analysis of cervical dysplasia and invasive cancer, there was another novel locus on chromosome 19p13 (rs425787) near *CD70* [122]. This large GWAS had not yet been fully peer-reviewed at the time of this review but will increase the number of potential cervical cancer risk loci and candidate genes that merit validation.

In a genome-wide sequencing study from Iceland, rare loss-of-function variants in *PTPN14* were associated with high risk of cervical cancer (OR, 12.7, *p* = 1.6 × 10^−4^) [123]. These overall variants were also associated with earlier age at diagnosis. *PTPN14* is a p53-regulated gene that encodes a protein tyrosine phosphatase targeted by high-risk HPV E7 proteins, and the authors speculate that its germline mutation may enhance oncogenic activation of the Hippo–YAP pathway [123]. This finding indicates a potential high-penetrance germline predisposition for cervical cancer, however, the role of PTPN14 needs replication in other populations and corroboration in functional studies.

### 2.4. Other GWAS Loci for Viral Infection

Since not all women infected with high-risk HPV (hrHPV) go on to develop cancer and it is known that viral diversity correlates with an excess of genomic variants in immune response loci [155], there has been obvious interest to understand genetic factors that contribute to a persistent HPV infection. Such an attempt to identify host genetic factors that influence HPV seropositivity found a variant at 6p21.32, rs9357152, to be associated with seropositivity for HPV8, a β-cutaneous HPV type [156]. Although this is not among the high-risk types for cervical cancer, the result provided supportive evidence that genomic variants in the HLA region can influence immune response against HPV. Another GWAS in a Nigerian population reported further variants associated with cervical hrHPV infections at sub-genome-wide significance [157]. Increasing study sizes in the future may allow genomic factors associated with viral exposure to be established that could help to identify women at increased risk at an early stage [158].

## 3. Follow-Up Studies of Cervical Cancer GWAS Results

As mentioned in the previous sections, the identification of significantly associated variants is just the first step. Apart from GWAS, gene-based and pathway analysis thus contribute towards the understanding of cervical cancer aetiology. At present, the genes underlying cervical cancer GWAS signals are largely unknown, although prediction tools are established to prioritise genes by the use of established data on large-scale chromatin conformation or tissue-specific gene expression. A transcriptome-wide association study (TWAS) based on the GWA studies by Leo et al. [66] and Takeuchi et al. [119] identified 20 genes to be associated with cervical cancer in using transcriptome databases for six different tissues [159]. These genes were mainly expressed at the HLA locus, however, four non-HLA genes were also identified. However, the tissues used in this study did not include cervical epithelial cells or cervical cancer lines, and these findings need further replication [159].

There are well-characterised methylation changes in cervical cancer prognosis, and an integrative analysis combining multi-omics approaches may help to further assign functional roles to susceptibility variants and understand the mechanisms underlying cervical cancer. Recent multi-omics approaches in tumours found that HPV related squamous carcinomas have defined molecular and genetic signatures [160]. However, the genomic germline factors determining hereditary cervical cancer risk and the somatic epigenetic and genetic variations do not necessarily share a large overlap. Nevertheless, the integration of methylome, proteome, and metabolome data could aid to narrow down causal genes and eventually identify novel risk factors.

While these processes of gene identification and functional follow-up are ongoing, parallel work will aim to make use of the identified genomic risk factors to define the individual risk of cancer in an unaffected woman with higher precision. Biobank-based large cohorts provide the possibility of testing the correlation between traits and draw polygenic risk scores (PRS) that can eventually help to design preventive measures and personalise treatment methods. In correlation studies, cervical cancer was not strongly correlated with other gynaecological cancers [68], although it has been found to be correlated with bladder cancer in one analysis [112]. In attempts to define polygenic risk scores, it has not been possible thus far to predict a strong PRS for cervical cancer due to the low number of known susceptibility variants given as input [161]. Nevertheless, polygenic risk scores can be a powerful instrument when more genomic risk loci become identified, as was shown for breast cancer [162], and this also bears great potential for cervical cancer [163]. 

In addition, Mendelian randomisation studies can be very useful for the robust identification of associated traits and will become more powerful with the increasing size of cervical cancer GWAS data. In this type of analysis, genetic variants replace exposure measures as instrumental variables to infer whether a risk factor affects a clinical outcome. The assumption is that the genetic variant is associated with the risk factor and influences the outcome only through the risk factor, independent of confounders. Thus far, Mendelian randomisation studies did not detect causal relationships between cervical cancer and obesity [164] or cervical cancer and Alzheimer’s disease [165], but they suggested a possible link between cervical cancer and type II diabetes mellitus [166], and they strongly supported the complementary role of smoking, older age at first pregnancy, and number of sexual partners in the risk of developing cervical cancer [121].

## 4. Conclusions

There is now ample evidence for a strong genetic component of cervical cancer risk that adds to known risks posed by epidemiological factors and HPV infection. There have been few cervical cancer GWA studies so far, with the HLA region emerging as a consistently associated locus across different populations. Cervical cancer thus adds to the large number of autoimmune and immune-mediated diseases associated with this genetic region [127,167,168]. As for most other HLA-associated cancers with infectious aetiology, the associations were driven by multiple HLA regions, suggesting that both cytotoxic and helper T-cell responses might be important [169]. This may open a perspective for personalised antigen-specific disease prevention through harnessing HLA–ligand interactions for clinical benefit [127]. More recent biobanking and consortia efforts have also led to the identification of non-HLA susceptibility loci such as rs8067378 (*GSDMB*), rs10175462 (*PAX8*), and rs27069 (*CLPTM1L-TERT*), which warrant further investigation. Such association results from large-scale case-control series will become increasingly useful to inform risk assessment strategies as well as biological and pharmacological studies.

Cohort size, apart from effect allele frequency and the magnitude of risk, determines the power of the study and is therefore a crucial factor in identifying novel susceptibility loci. There are only a limited number of consistent cervical cancer susceptibility variants thus far, compared to other traits, and even fewer follow-up studies to identify the causal variant via fine-mapping. Even where the causal variants were determined, it still remains a challenge to assign biological function to the variants and identify the causal gene(s). Furthermore, only a handful of results so far has been attempted for fine-mapping or functional validation via chromatin conformation or eQTL analysis [119,128,132,142,152,170].

Although there are rare coding variants in *MICA*, *HLA-B*, or *PTPN14* that would directly affect the function of the gene product [121,123,128], the majority of genome-wide significant signals identified at susceptibility regions seem to be regulatory variants. For the HLA region, specific amino acids in the MHC-peptide binding groove may also explain some of the SNP associations, as is known for *HLA-B* and HIV infection [171,172]. Additional work will be required to trace down target genes at every locus and to understand their molecular mechanisms. It is highly likely that at least some of these genomic factors play a role in viral infection and host immunity. The identification of these genes and their function(s) could provide important insights into the aetiology of cervical cancer and fuel further pharmacological research [173].

Thus far, the low number of known cervical cancer susceptibility variants and genes has restricted further research towards precision medicine approaches. It is therefore the need of the hour to perform large-cohort based cervical cancer GWAS and meta-analysis in order to identify further susceptibility loci, and national biobanking has proven to be a very fruitful resource for genomic research in this regard. With ongoing efforts, we may eventually be able to screen for disease-associated risk factors early on as part of preventive cancer surveillance strategies.

## Figures and Tables

**Figure 1 cancers-13-05137-f001:**
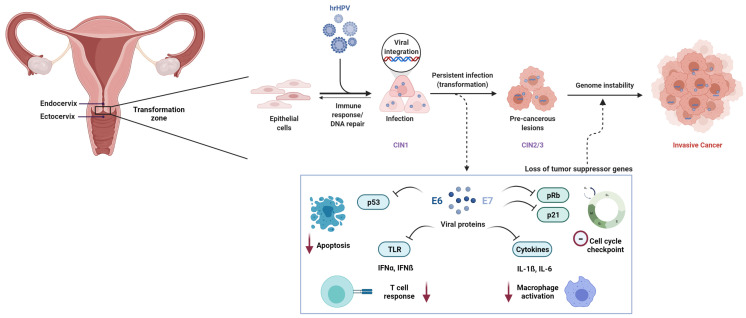
Pathogenesis of cervical cancer. Epithelial cells in the transformation zone of the cervix acquire lesions upon persistent infection with high risk HPV (hrHPV). In some cases, the lesions resolve, whereas in others, upon viral integration, cells are transformed and progress from cervical intraepithelial neoplasia I to II and III (CIN1, CIN2, and CIN3). Viral proteins E6 and E7 are released and inhibit apoptosis mediated by TP53, cell cycle checkpoint by p21, T-cell response by toll-like receptors (TLR), and macrophage activation by cytokines. This leads to an insufficient immune response and viral replication, uncontrolled cell proliferation and genome instability, and further cancer in situ (CIS) or invasive cervical cancer (CC).

**Figure 2 cancers-13-05137-f002:**
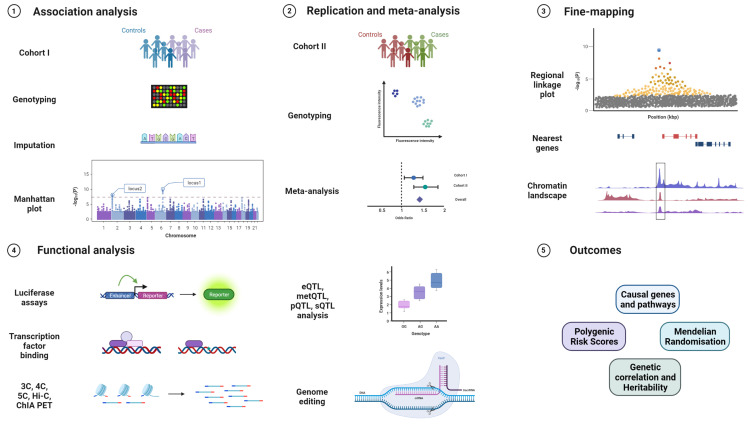
GWAS workflow from replication, validation, fine-mapping, and identifying biological mechanisms to clinically relevant outcomes. The various stages of a genome-wide association study, starting from genotyping on custom arrays, imputation on reference genomes, association analysis, and visualisation, followed by replication in an independent cohort, validation genotyping, and meta-analysis. The top loci are then fine-mapped and integrated with bioinformatic annotations before proceeding to functional experiments in relevant cell and tissue types such as promoter and enhancer luciferase assays, ChIP, 3C, 4C, 5C, Hi-C, ChIA-PET, eQTL analysis, and genome editing via the CRISPR/Cas system. Expected outcomes are the identification of relevant genes and pathways affected by the variant, and extraction of polygenic risk scores (PRS), Mendelian randomisation (MR), and genetic correlation with other traits.

**Table 1 cancers-13-05137-t001:** List of cervical cancer GWAS performed so far, with study population, genome-wide significant variants, and references. Signals that are correlated at r^2^ > 0.3 [110] with other GWAS variants listed above are indicated with a–e. Replication results are indicated with an asterisk (*) before the rsID. LoF: loss of function.

GWAS	Population	GWS Risk Loci		Comment	Ref.
Chen et al., 2013	Swedish	6p21.33	rs2516448 (*MICA*)		[116]
		6p21.32	rs9272143 (*HLA-DRB1*/*HLA-DQA1*) ^a^		
		6p21.32	rs3117027 (*HLA-DPB2*)		
Shi et al., 2013	Chinese	6p21.32	rs4282438 (*HLA-DPB1*/*HLA-DPB2*)		[117]
		4q12	rs13117307 (*EXOC1*)		
		17q12	rs8067378 (*GSDMB*) ^b^		
Chen et al., 2016	Swedish	6p21.32	* rs9271898 (*HLA-DQA1*) ^a^	Pooled analysis with mainly CIN3	[118]
		6p21.33	* rs2516448 (*MICA*)		
		6p21.32	* rs3130196 (*HLA-DPA2*)		
		6p21.32	rs73730372 (*HLA-DQA1/HLA-DQB1*) ^c^		
		6p21.32–6p21.33	HLA alleles *HLA-B** 07:02, *HLA-B** 15:01, *HLA-DRB1** 13:01, *HLA-DRB1** 15:01, *HLA-DQA1** 01:03, *HLA-DQB1** 06:03, *HLA-DQB1** 06:02, *HLA-C** 07:02		
Leo et al., 2017	Caucasian	6p21.32–6p21.33	Replication of HLA haplotypes that are determined by the amino-acids carried at positions 13 and 71 of HLA-DRB1 and position 156 in HLA-B		[66]
Rashkin et al., 2020	UK/US	2q13	rs10175462 (*PAX8/PAX8-AS1*) ^d^	Combined analysis of UK and GERA biobanks	[68]
		6p21.32	rs2856437 (*PBX2*)		
Takeuchi et al., 2019	Japanese	5q14.3	rs59661306 (*ADGRV1/ARRDC3*)		[119]
		7p11.2	rs7457728 (*LINC01445/VSTM2A*)		
Ishigaki et al., 2020	Japanese	no new loci for cervical cancer			[120]
Bowden et al., 2021	UK/Finnish	2q13	* rs35724515 (*PAX8/PAX8-AS1*) ^d^	Analysis of UK Biobank and validation in the FinnGen biobank.	[121]
		5p15.3	rs27069 (*CLPTM1L*)		
		6p21.32	* rs9272050 (*HLA-DQA1*) ^a^		
		6p21.33	rs6938453 (*MICA*)		
		6p21.32	* rs55986091 (*HLA-DQB1*) ^c^		
		6p21.33	rs9266183 (*HLA-B*)		
		6p21.32	* rs9272245 (*HLA-DQA1*) ^a^		
		1p36.32	rs138446575 (*TTC34*)		
		12q24.11	rs117960705 (*ACACB*)		
Koel et al., 2021	Multi- ethnic	1p36.12	rs2268177 (*LINC00339/CDC42*)	Meta-analysis of UK, FinnGen, Japanese RIKEN and Estonian biobanks.	[122]
		2q13	* rs4849177 (*PAX8/PAX8-AS1*) ^d^		
		5p15.3	* rs27069 (*CLPTM1L*)		
		17q12	* rs12603332 (*GSDMB*) ^b^		
		2q24.1	rs12611652 (*DAPL1*)		
		6p21.32	* rs35508382 (*HLA-DRB1/HLA-DQA1*) ^c^		
		6p21.33	rs1053726 (*HLA-B*) ^e^		
		6p21.32	* rs36214159 (*HLA-DQA1*) ^c^		
		19p13.3	rs425787 (*CD70*)		
** *Gene-based analysis* **					
Olafsdottir et al., 2021	Icelandic	1q32.3–41	3 LoF variants (c.-1_2delinsATGG, p.Gln503ArgfsTer12, c.3271 + 1G > A) in *PTPN14*	Burden analysis after imputation of rare PVs	[123]
** *Cross-trait analysis* **					
Masuda et al., 2019	Japanese	11p15.5	rs150806792 (*INS-IGF2*)	Combined analysis of cervical cancer with uterine cancer (*INS-IGF2, SOX9)* or five different cancers	[124]
		17q24.3	rs140991990 (*SOX9*)		
		2p16.3	rs937380553 (*LOC730100*)		
		9q22.33	rs73494486 (*GABBR2*)		
		15q25.2	rs145152209 (*SH3GL3/BNC1*)		
		21q22.2	rs147427629 (*LOC107985484*)		
Rashkin et al., 2020	UK/US	4q24	rs10007915 (*TET2*)	Combined analysis of cervical cancer with 15 different cancers	[68]
		6p21.33	rs17190106 (*MUC22/HCG22*)		
		6p21.33	*rs9266766 (*HLA-S/MICA*) ^e^		
		6p21.33	rs114060326 (*MICB/MCCD1*)		
		6p21.33	rs2763979 (*HSPA1B*)		
		6p21.32	rs34563311 (*HLA-DRB1*)		
		6p21.32	* rs9270747 (*HLA-DRB1/HLA-DQA1*) ^a^		
		6p21.32	rs535777 (*HLA-DRB1/HLA-DQA1*)		
		8q24.21	rs117952826 (*CASC8*)		

## Data Availability

The GWAS summary statistics for most of the studies described in this text are available from the following online repositories, in addition to the respective cited research articles. Leo et al. (https://www.ebi.ac.uk/gwas/efotraits/EFO_0001061GWASCatalog, Accession ID GCST004833), Rashkin et al. (https://www.ebi.ac.uk/gwas/efotraits/EFO_0001061GWASCatalog, Accession ID GCST90011816), UK Biobank (CC GWAS with female controls only, https://github.com/Nealelab/UK_Biobank_GWAS, file: 20001_1041.gwas.imputed_v3.female), and FinnGen freeze 5 (https://r5.finngen.fi/), Japan Biobank (https://pheweb.jp/).

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
