# Peer review of "Genomic Risk Factors for Cervical Cancer"

_cancers, 2021, doi:10.3390/cancers13205137_

Round 1
Reviewer 1 Report
This is a well-written paper by Dhanya Ramachandran and Thilo Dörk summarizing findings from recent genome wide association studies (GWAS) on cervical cancer and proposing variants which may be universal susceptibility loci.
Authors reviewed cervical cancer susceptibility variants arising from recent genome-wide association studies and meta-analysis in large cohorts and proposed three consistently replicated non-HLA cervical cancer susceptibility loci as follows: 2q14 (PAX8), 17q12 (GSDMB), and 5p15.33 (CLPTM1L). They discussed the available evidence for these loci knowledge gaps, future perspectives, and the potential impact of these findings on precision medicine strategies to combat cervical cancer. The authors also extensively discussed latest literature data concerning HLA cervical cancer susceptibility loci.
This is an excellent study, methodologically and clinically valuable. The study was well performed and the data were thoroughly analyzed and interpreted. The authors extensively discussed presented data.
Paper contains 1 table summarizing the latest results from cervical cancer GWAS and two schemes illustrating cervical cancerogenesis and GWAS workflow. The latter was very helpful and informative. The list of references comprises 174 papers.
The manuscript includes comprehensive information on epidemiology, HPV-associated pathogenesis and heritability of cervical cancer.
Minor comments:
Comment 1. Few papers may be outdated, thus for some readers these full papers might be hardly available: Bender, S. Carcinoma in-situ of cervix in sisters. BMJ 1976, 1, 502–502, doi:10.1136/bmj.1.6008.502. 674; Bruinse, H.W.; te Velde, E.R.; de Gast, B.C. Human leukocyte antigen patterns in a family with cervical cancer. Gynecol. Oncol. 678 1981, 12, 249–252, doi:10.1016/0090-8258(81)90154-2.
Comment 2. The title of the manuscript may be somehow misleading as authors focus mostly on non HLA cervical cancer susceptibility loci; 2q14 (PAX8), 17q12 (GSDMB), and 5p15.33 (CLPTM1L). Nevertheless, HLA region as the first consistent cervical cancer susceptibility locus was also thoroughly discussed in this paper.
Taken together, this paper by Dhanya Ramachandran and Thilo Dörk represents a worthwhile contribution to the cancer research. I recommend the manuscript for further publication process.
Author Response
Comment 1. Few papers may be outdated, thus for some readers these full papers might be hardly available: Bender, S. Carcinoma in-situ of cervix in sisters. BMJ 1976, 1, 502–502, doi:10.1136/bmj.1.6008.502. 674; Bruinse, H.W.; te Velde, E.R.; de Gast, B.C. Human leukocyte antigen patterns in a family with cervical cancer. Gynecol. Oncol. 678 1981, 12, 249–252, doi:10.1016/0090-8258(81)90154-2.
Authors´ reply: We agree that these are old references but we felt that, in a comprehensive review, we should also give credit to those who were the first in the field to report some evidence of the heritability of cervical cancer. However, during revision of this manuscript, we came across a much earlier report of three sisters with cervical cancer that had been published in Lancet (Way S, Hetherington J, Galloway DC. doi: 10.1016/s0140-6736(59)90810-4. PMID: 13843203). Because Lancet is a journal more accessible to many readers, we have substituted Bender et al. with this new reference and added a short sentence to pinpoint this historical aspect.
Comment 2. The title of the manuscript may be somehow misleading as authors focus mostly on non HLA cervical cancer susceptibility loci; 2q14 (PAX8), 17q12 (GSDMB), and 5p15.33 (CLPTM1L). Nevertheless, HLA region as the first consistent cervical cancer susceptibility locus was also thoroughly discussed in this paper.
Authors´ reply: We agree that it is important to cover both the HLA and non-HLA genomic loci in such a review. Our title “Genomic risk factors for cervical cancer” was thought to reflect this broader scope. The non-HLA regions have become apparent only recently and have not been reviewed previously, therefore these variants deserve to be discussed in some depth. On the other hand, the HLA variants are many and represent a complex region that is also important to work out in some detail. Our review contains some 1000 words on HLA variants and some 1100 words on non-HLA variants, thus it should hopefully provide a balanced overview.
Taken together, this paper by Dhanya Ramachandran and Thilo Dörk represents a worthwhile contribution to the cancer research. I recommend the manuscript for further publication process.
Authors´ reply: Thank you for your recommendation and your time in evaluating our manuscript.
Reviewer 2 Report
In this review, Ramachandran & Dörk discuss the current knowledge of genomic risk factors and the molecular alterations found in cervical cancer. It is well-written, well-structured, needed, and shows a useful overview of cervical cancer. Overall the review presents the reader with an easy and entire overview of the current research status of this interesting topic.
Optional:
A table of the used abbreviations at the beginning of the text might be useful for the reader.
Author Response
In this review, Ramachandran & Dörk discuss the current knowledge of genomic risk factors and the molecular alterations found in cervical cancer. It is well-written, well-structured, needed, and shows a useful overview of cervical cancer. Overall the review presents the reader with an easy and entire overview of the current research status of this interesting topic.
Authors´ reply: Thank you for your expert comments and your time in evaluating our manuscript.
Optional:
A table of the used abbreviations at the beginning of the text might be useful for the reader.
Authors´ reply: Thank you for your suggestion. In the revised version, we now have placed a short list of abbreviations before the text.